# WaveFake: A Data Set to Facilitate Audio DeepFake Detection

**Joel Frank**[*]
Ruhr University Bochum
Horst Görtz Institute for IT-Security

**Lea Schönherr**
Ruhr University Bochum
Horst Görtz Institute for IT-Security

## Abstract

Deep generative modeling has the potential to cause significant harm to society. Recognizing this threat, a magnitude of research into detecting so-called "Deep-Fakes" has emerged. This research most often focuses on the image domain, while studies exploring generated audio signals have—so-far—been neglected. In this paper we make three key contributions to narrow this gap. First, we provide researchers with an introduction to common signal processing techniques used for analyzing audio signals. Second, we present a novel data set, for which we collected audio samples from five different network architectures, across two languages. Finally, we supply practitioners with two baseline models, adopted from the signal processing community, to facilitate further research in this area.

## 1 Introduction

$243,000 were lost, when criminals used a generated voice recording to impersonate the CEO of a UK company [76]. This is just one of several reports where current state-of-the-art generative modeling was used in harmful ways. Other examples include: impersonation attempts [20], influencing opposition movements [36], being used to justify military actions [24, 46], or online harassment [9]. While there is a multitude of beneficial use cases, for example, enhancing data sets for medical diagnostics [18, 22], medical image segmentation [87], or designing DNA to optimize protein bindings [29], finding effective ways to detect fraudulent usage is of utmost importance to society.

Detection in the image domain has received tremendous attention [41, 45, 91, 78, 83, 43, 47, 42, 17, 21]. However, the audio domain is severely lacking. While there does exist prior work exploring image and sound together (i.e., videos) [13], an analysis of audio in isolation is missing. This is a critical gap. When examining the domains jointly, we can utilize synergies, for example, analyzing how well spoken audio matches video on screen.

To encourage more researchers to also explore the audio domain, we make three key contributions in this paper: First, we provide an overview of common signal processing techniques used for analyzing audio signals. We give an introduction to spectrograms, which are commonly used as an intermediate representation for generative models [35, 60, 88, 89], Additionally, we review common feature representations used for automatic speech recognition [56] or speaker verification [67], and provide a survey of current state-of-the-art generative models.

Second, our main contribution is a novel data set. We collected eight sample sets from five different network architectures across two languages. In this paper, we focus on analyzing samples which resemble (i.e., recreate) the training distributions. This allows for one-to-one comparisons of audio clips between the different architectures, in which we find subtle differences between the generators.

---

[*]Corresponding author `joel.frank@rub.de`.

Submitted to the 35th Conference on Neural Information Processing Systems (NeurIPS 2021) Track on Datasets and Benchmarks. Do not distribute.

Additionally, we expect good performing classifiers to transfer well to other contexts, since recreating the training distribution should yield the most quality samples.

Finally, we implement two classifiers, which we adopted from best practices in the signal processing community [67], to give future researchers a baseline to compare against [2]. Furthermore, we implemented BlurIG [86] a popular attribution methods, so practitioners can inspect their predictions when building on our results.

We summarize our main contributions as follows:

- An introduction into common signal processing techniques and a survey of the current landscape of audio generative modeling.
- A novel data set consisting of samples from several state-of-the-art generative network architectures.
- An implementation of two baseline models for future researchers to compare against.

## 2 Background

In this section we provide an introduction into common techniques used for analyzing speech audio signals. The list is far from exhaustive, but it provides a starting point for researchers to explore the field. The interested reader is refereed to the excellent books by Rabiner et al. [63] or Quatieri [62]. Additionally, we provide a survey on current state-of-the-art generative models and summarize related work.

### 2.1 Analyzing speech signals

We start by giving an introduction to commonly used techniques and representations used to analyze audio signals.

**(Mel) spectrograms:** A spectrogram is a visual representation of the frequency information of a signal over time (cf. Section 3, Figure 2 for an example). To calculate a spectrogram for an audio signal, we proceed as follows: First we divide the waveform into *frames* (e.g., 20 ms) with an overlap (e.g., 10 ms) between two adjacent frames. We then apply a window function $w(n)$ to avoid spectral leakage [3]. These functions (e.g., Hamming, Hann, Blackman window) are a trade-off between frequency resolution and spectral leakage and their choice depends on the task and the signal properties, cf. Prabhu [57] for a detailed overview. We multiply each individual frame from our audio signal with the windowing function:

$$x_w(t, n) = x(t, n) \cdot w(n) \quad \forall n = 0, \ldots, N - 1, \tag{1}$$

where $N$ is the frame length and $t = 1, \ldots, T$ the frame index of the signal sample $x(t, n)$. The frames are then transformed individually using the *Discrete Fourier Transform* (DFT) to obtain a representation in the frequency domain:

$$X(t, k) = \sum_{n=0}^{N-1} x_w(t, n) e^{-i2\pi \frac{kn}{N}} \quad \forall k = 0, \ldots, K - 1, \tag{2}$$

with $K$ DFT coefficients. This procedure of dividing the input signal, applying the window function and computing the DFT is refereed to as the *Short-Time Fourier Transform* (STFT). Finally, we calculate the squared magnitude $|X(t, k)|^2$ of the complex-valued signal to obtain our final representation—the spectrogram.

A commonly used variant is the so-called Mel spectrogram. It is motivated by studies which have shown that humans do not perceive frequencies on a linear scale. In particular, they can detect differences in lower frequencies on a more fine grade scale when compared to higher frequencies [97]. The Mel scale is an empirically determined non-linear transformation which approximates this relationship:

$$f_{\text{mel}} = 2595 \cdot \log_{10}\left(1 + \frac{f}{700}\right), \tag{3}$$

---

[2]Our code can be found at `github.com/RUB-SysSec/WaveFake`
[3]Energies from one frequency leak into other frequency bins.

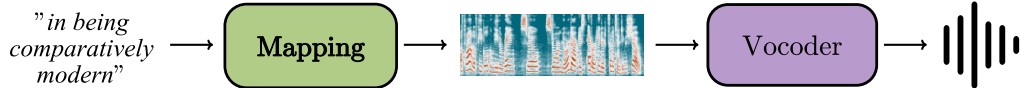

Figure 1: **A typical TTS pipeline.** One model takes a textual prompt with the desired audio transcription (we call it the "mapping" model) and outputs an intermediate representation, for example Mel spectrograms. This intermediate representation is then fed to a second model (in the literature often refereed to as "vocoder") to obtain the final raw audio.

where $f$ is the frequency in Hz and $f_{\mathrm{mel}}$ the Mel-scaled frequency. To obtain Mel spectrogram, we apply an ensemble of $S$ triangular filters $H_{\mathrm{mel}}$ (we provide a visual representation in the supplementary material). These filters have a linear distance between the triangle mid frequencies in the Mel scale, which results in a logarithmic increasing distance of the frequencies in the frequency domain

$$X_{\mathrm{mel}}(t,s) = \sum_{k=0}^{K-1} |X(t,k)| H_{\mathrm{mel}}(s,k) \quad \forall\, s = 1, \ldots, S. \tag{4}$$

Which gives us the final Mel spectrogram. Based on it, we can compute a common feature representation for audio analysis:

**Mel Frequency Cepstral Coefficients:** *Mel Frequency Cepstral Coefficients* (MFCC) are derived from a Mel-scaled spectrogram by applying a *Discrete Cosine Transform* (DCT) to the logarithm of the Mel-filtered signal

$$c(t,r) = \sum_{s=0}^{S-1} \log\left[X_{\mathrm{mel}}(t,s)\right] \cdot \cos\left[\frac{\pi \cdot r \cdot (s+0.5)}{S}\right] \quad \forall\, r = 0, \ldots, R-1, \tag{5}$$

where $R$ is the number of DCT coefficients.

**Linear Frequency Cepstral Coefficients:** We can also calculate *Linear Frequency Cepstral Coefficients* (LFCC). As the name suggest these coefficients are derived by applying a linear filterbank (instead of a Mel filterbank) to the spectrogram of the signal. This results in retaining more high frequency information. Except for the replacement of the filter bank, all other step remain the same as for MFCC features.

**(Double) delta:** MFCCs and LFCCs are often augmented by their first and second derivatives to represent temporal structure of the input. These are refereed to as delta and double delta features, respectively. In practice these are often calculated by central difference approximation via

$$d(t) = \frac{\sum_{n=1}^{N} n \cdot \left[c\left(t+n\right) - c\left(t-n\right)\right]}{2 \cdot \sum_{n=1}^{N} n^2} \quad \forall\, t = 0, \ldots, T-1, \tag{6}$$

where $d(t)$ is the delta at time $t$ and $N$ is a user-defined window length for computing the delta, and $c$ is either the MFCCs/LFCCs or the delta features (when calculating the double delta features).

## 2.2 Text-to-speech (TTS)

In this Section we want to give a broad overview over different research direction for *Text-To-Speech* (TTS) models. Due to the rapid developments of the field, this is a non-exhaustive list. However, it serves as a starting point for interested researchers.

While there has been some research into end-to-end models [16, 77], typical TTS models consist of a two-stage approach, represented in Figure 1. First, we enter the text sequence which we want to generate. This sequence gets mapped by some model (or feature extraction method) to a low-dimensional intermediate representation, often linguistic features [7] or Mel spectrograms [49]. Second, we use an additional model (often refereed to as vocoder), to map this intermediate representation to raw audio. We focus on the literature on vocoders, since it directly connects to our work.

Today, vocoders are typically implemented by Deep Neural Networks (DNNs). The first DNN [93, 19] approaches adopted the parametric vocoders of earlier HMM-based models [94, 80, 90]. Here the DNN was used to predict the statistics of a given time frame, which are then used in traditional speech parameter generation algorithms [80]. Later variants replaced each component in traditional pipelines with neural equivalents [7, 6, 64, 65, 84, 4]. The first architectures who started using DNNs exclusively as the vocoder were auto-regressive generative models, such as WaveNet [49], WaveRNN [27], SampleRNN [44], Char2Wav [75] or Tactron 2 [72].

Due to their auto-regressive nature, these models do not leverage the parallel structure of modern hardware. There have been several attempts to circumvent this problem: One direction is to distill trained auto-regressive decoders into flow-based [32] convolutional student networks, as done by Parallel WaveNet [49] and Clarinet [54]. Another method is to utilize direct maximum likelihood training as done by several flow-based models, for example, WaveGlow [60], FloWaveNet [30] or WaveFlow [55]. Other probabilistic approaches include those based on variational auto-encoders [50, 53] or diffusion probabilistic models [34, 12]. Another family of methods is based on Generative Adversarial Networks (GANs) [23], examples include, MelGAN [35], GAN-TTS [8], WaveGAN [15], Parallel WaveGAN [88] or Multi-Band MelGAN [89].

## 2.3 Related Work

There have been several previous proposals which collected DeepFake data: The FaceForensics++ dataset [66] curated 1.8 million manipulated images and provides a benchmark for automated facial manipulation detection. Celeb-DF [40] contains high-quality face-swapping DeepFake videos of celebrities with more than 5,000 fake videos. Dolhansky et al. [14] released the DeepFake detection challenge that contains more than 100,000 videos, generated with different methods.

There exists a multitude of research into identifying GAN-generated images: Several approaches use CNNs in the image domain [41, 45, 91, 78, 83], others use statistics in the image domain [43, 47]. Another group of systems employs handcrafted features from the frequency domain: steganalysis-based features [42], spectral centroids [82] or frequency analysis [96, 17, 21, 61]. Li and Lyu [39] proposed a CNN-based DeepFake video detection framework which utilizes artefacts that are consequences of the generation process. Another strain of research combines image analysis with audio analysis. Chintha et al. [13] combined a DeepFake detection with an audio spoofing detection to identify fake videos. At the time of writing and to the best of our knowledge no work has analyzed DeepFake audio in isolation.

A related line of research is undertaken by the signal processing community. The biyearly ASVspoof challenges [85, 79, 48] promotes countermeasure against spoofing attacks that aim to fool speaker verification systems via different kinds of attacks. Their benchmarking data sets include replay attacks, voice conversion and synthesized audio files. Note that the 2021 edition of the challenge features an audio DeepFake track, but does not provide specific training data for it. We imagine our data set to be used complementary with the training data of the challenge. At the time of writing the 2021 edition is still on-going, but evaluating the best performing models in conjunction with our data set is an interesting direction for future work. In the mean time, we adopt one of the baseline models of the ASVspoof challenge to enable a direct comparison. These efforts have lead to several proposed models for detecting spoofing attacks, for example, CNN-based methods [81, 38, 37], ensemble methods on different feature representations [52, 28, 69] or methods which detect unusual pauses in human speech [95, 3]. Additionally, another data set is proposed by Kinnunen et al. [33]. They released a re-recorded version of the RedDots database for replay attack detection text-dependent speaker verification.

## 3 The data set

In this Section we provide an overview of our data set. It consists of 88,600 generated audio clips (16-bit PCM wav) and can be found on zenodo [4]. We mostly base our work on the LJSPEECH [26] data set. While TTS models often get trained on private data sets, LJSPEECH is the most common public data set among the publication listed in Section 2.2. Additionally, we consider the JSUT [74] data set, a Japanese speech corpus.

---

[4] `zenodo.org/record/4904579` - DOI: 10.5281/zenodo.4904579

**Reference data:** We examine multiple networks trained on two reference data sets. First, the LJSPEECH [26] data set consisting of 13,100 short audio clips (on average 6 seconds each; roughly 24 hours total) read by a female speaker. It features passages from 7 non-fiction books and the audio was recorded on a MacBook Pro microphone. Second, we include samples based on the JSUT [74] data set, specifically, basic5000 corpus. This corpus consists of 5,000 sentences covering all basic kanji of the Japanese language (4.8 seconds on average; roughly 6.7 hours total). The recordings were performed by a female native Japanese speaker in an anechoic room. Thus, our data set consists of approximately 157 hours of generated audio files in total. Note that we do not redistribute the reference data. They are freely available online [26, 74].

**Architectures:** We included a range of architectures in our data set:

- **MelGAN**: We include MelGAN [35], which is one of the first GAN-based generative models for audio data. It uses fully convolutional feed-forward network as generator and operates on Mel spectrograms. Their discriminator is a combination of three different discriminators that operates on the original, and two downsampled versions of the raw audio input. Additionally, they use an auxiliary loss over the feature space of the three discriminators.

- **Parallel WaveGAN (PWG)**: WaveNet [49] is one of the earliest and most common architectures, We include samples from one of its variants, Parallel WaveGAN [88]. It uses GAN training paradigm, with a non-autoregressive version of WaveNet as its generator. In a similar vein to MelGAN, it uses an auxiliary loss, but in contrast, matches the STFT of the original training sample and the generated waveform over mutliple resolutions.

- **Multi-band MelGAN (MB-MelGAN)**: Incorporating more fine-grained frequency analysis, might lead to more convincing samples. We include MB-MelGAN, which computes its auxiliary (frequency-based; inspired by PWG) loss in different sub-bands. Its generator is based on a bigger version of the MelGAN generator, but instead of predicting the entire audio directly, the generator produces multiple sub-bands, which are then summed up to the complete audio signal.

- **Full-band MelGAN (FB-MelGAN)**: We include a variant of MB-MelGAN which generates the complete audio directly and computes its auxiliary loss (the same as PWG) over the full audio instead of its sub-bands.

- **WaveGlow**: The training procedure might also influence the detectability of fake samples. Therefore, we include samples from WaveGlow to investigate maximum-likelihood-based methods. It is a flow-based generative model based on Glow [31], whose architecture is heavily inspired by WaveNet.

Additionally, we examine MelGAN both in a version similar to the original publication, which we denote as MelGAN, and in a larger version with a bigger receptive field, MelGAN (L)arge. This version is similar to the one used by FB-MelGAN, allowing for a one-to-one comparison. In total, we sample eight different data sets, six based on LJSPEECH (MelGAN, MelGAN (L), FB-MelGAN, WaveGlow, PWG) and two based on JSUT (MB-MelGAN, PWG).

**Sampling procedure:** For WaveGlow we utilize the official implementation [59] (commit 8afb643) in conjunction with the official pre-trained network on PyTorch Hub [58]. We use a popular implementation available on GitHub [25] (commit 12c677e) for the remaining networks. The repository also offers pre-trained models. When sampling the data set, we first extract Mel spectrograms from the original audio files, using the pre-processing scripts of the corresponding repositories. We then feed these Mel spectrograms to the respective models to obtain the data set. Intuitively, the networks are asked to "recreate" the original data sets.

**Differences between the architectures:** We analyze differences between the architectures by plotting the spectrograms of an audio file in Figure 2 (LJSPEECH 008-0217). Larger plots can be found in the supplementary material. Generally, all architectures produce spectrograms different to the original. The networks seem to generally struggle with the absent of information (solid circles in Figure 2a). They also seem to consistently produce differing results in the higher frequency, especially for vocals (dashed circle). Additionally, MelGAN and WaveGlow seem to cause a repeating horizontal pattern. The remaining networks (all using an auxiliary loss in the frequency domain) do not seem to

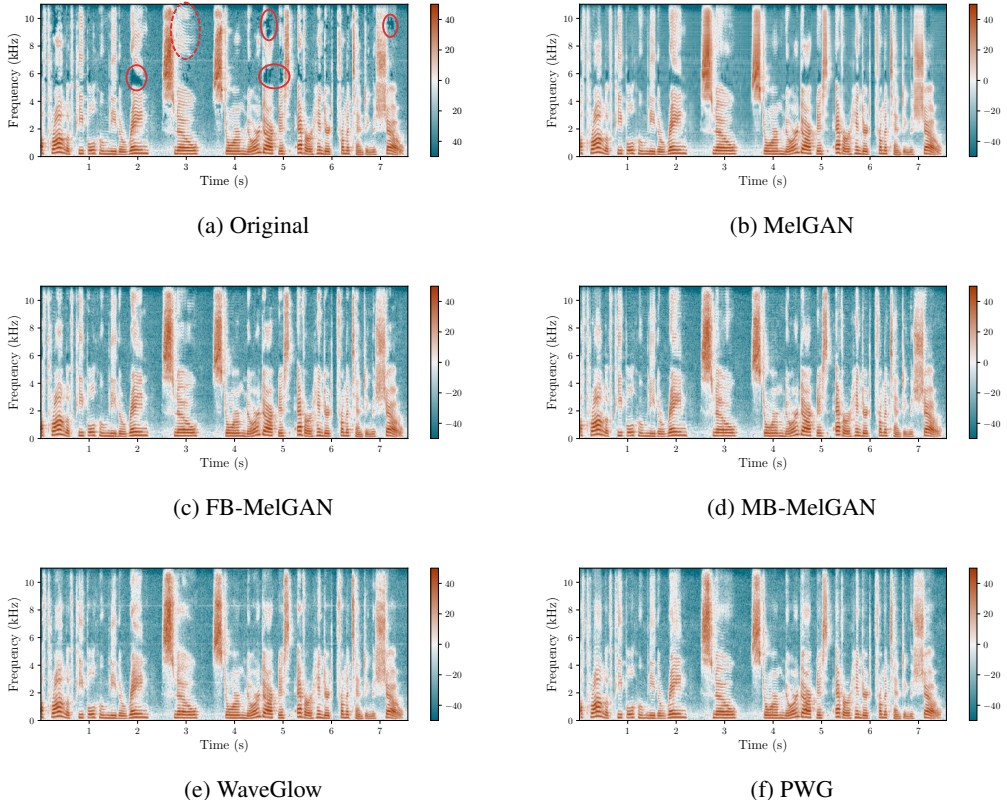

(a) Original            (b) MelGAN

(c) FB-MelGAN          (d) MB-MelGAN

(e) WaveGlow           (f) PWG

Figure 2: **Spectrograms for the same sample, for different generating models.** They show the frequencies of a signal, plotted over the time of a signal. Lower frequencies at the bottom, higher at the top. Best viewed in color.

exhibit this behaviour. However, they still produce clear differences. Note that these differences are visible when plotting the audio but generally inaudible when listening to the samples.

**A note on licensing:** During the collection of our data set, we ran into an interesting questions which we could not find a satisfying answer to. We generated samples which are intrinsically designed to be as close as possible to the original data set. So, when distributing these samples (or the models that generated them), it is not clear whether the original license does still apply. The data is obviously not the original data. Yet, it sounds remarkably like it. To the best of our knowledge this question has not been addressed by the machine learning or legal community.

For our sake, the LJSPEECHdata set is in the public domain. The JSUTcorpus is licensed by CC-BY-SA 4.0, with a note that redistribution is only permitted in certain cases. We contacted the author, who saw no conflict in distributing our fake samples, as long as its for research purposes.

To comply with JSUT we license our data set under the CC-BY-SA 4.0 license.

**Ethical considerations:** Our data set consists of phrases from non-fiction books (LJSPEECH) and every-day conversational Japanese (JSUT), which are already available online. The same is true for all models used to generate this data set.

One might wonder if releasing research into detecting DeepFakes might contribute negatively towards the detection "arms race". This is a long standing debate in the security community and the overall consensus it that "security through obscurity" does not work. Intuitively, withholding information from the research community is in-fact more harmful, since attackers will eventually adapt to any defense one deploys. We have provided a more thorough discussion of this topic in the supplementary material and we hope that this examination contributes to the overall dialogue on security analysis of machine learning systems.

# 4 Providing a baseline

To provide a point of reference for future researchers, we adopt the baseline model of the ASVspoof challenge [79]. A bi-yearly challenge on detecting spoofed audio samples.

## 4.1 Experimental setup

We start by training six different classifiers, one for each vocoder in our data set (MelGAN, MelGAN (L), FB-MelGAN, MB-MelGAN, PWG and WaveGlow). For training our classifiers, we exclusively use the data sets based on LJSPEECH. Additionally, we use the JSUT data as a hold-out set for accessing the classifiers ability to generalize to an unknown setting (different speaker, language, and recording setup). While we do not explicitly asses completely novel phrases, the JSUT experiments give us a good approximation. We follow Sahidullah et al. [67] and train two Gaussian Mixture Models (GMMs), one fitting the real distribution (the original LJSPEECH data set) and one fitting the generated audio samples (the respective vocoder-samples from our data set). In addition to the LFCC features used by Sahidullah et al. [67], we evaluate MFCC features, since they are a commonly used feature representation for audio tasks. We calculate the likelihood $\Lambda(\mathbf{X})$ of a test sample via

$$\Lambda(\mathbf{X}) = \log p(\mathbf{X}|\theta_n) - \log p(\mathbf{X}|\theta_s), \tag{7}$$

where $\mathbf{X}$ are the input features (namely MFCC or LFCC) and $\theta_n$ and $\theta_s$ are the GMM model parameter of the real and the generated audio distributions, respectively.

For each classifier we evaluate the performance on all vocoders over a hold-out set of 20% of the data. We use the *Equal Error Rate* (EER) as our evaluation metric. This metric is also been used by the ASVspoof challenge. It is defined as the point on the ROC curve, where false acceptance rate and false rejection rate are equal and is commonly used to assess the performance of binary classifications tasks like biometric security systems [68]. The best possible value is 0.0 (no wrong predictions), worst 1.0 (everything wrong), guessing is 0.5. The lower the EER the better the system performs. Additionally, we compute average EER over all test sets.

Finally, we train six additional models in a leave-one-out setting to access if the models picked up on vocoder-specific characteristics when trained on data produced by only one model. These models are exclusively trained on LFCC features.

**Training details:** We train the GMMs using the *Expectation Maximization* (EM) algorithm on 1,000 samples for a maximum of 100 iterations (the models generally converge after approximately 60 iterations), we use 128 mixture components and learn the diagonal covariance matrix of each distribution. To ensure we do not get stuck in a local minima, we randomly reinitialized the EM algorithm 10 times, picking the model with the highest log likelihood on the training data. We also trained GMMs using gradient descent on a larger training corpora ($\sim 10,000$ audio samples), to control for the size of our training set. The EM version obtained strictly better results. Training EM-based models for the leave-on-out experiments proved difficult due to numerical instability. Thus, we exclusively rely on gradient descent based models. We doubled the amount of mixture components (256) and epochs (20) to compensate for the more difficult task of fitting a more diverse training set.

We resample all audio files to 16kHz and remove silence parts which are longer than two seconds. When converting the audio files to MFCC/LFCC features, we use the parameters proposed by Sahidullah et al. [67]. We extract 20 LFCC/MFCC features and compute delta-/double-delta-features, cf.Section 2.1.

We trained all our models on a machine running Ubuntu 18.04.5 LTS, with a AMD Ryzen 7 3700X 8-Core Processor and 64GB of RAM. The implementation of our models was performed in PyTorch 1.8.1, using the torchaudio extension in version 0.8.1 [51]. The EM version of the GMM models can be trained exclusively on the CPU, taking roughly two and a half hours to train a single model (100 iterations; 10 reruns). When training the gradient descent version, we used a GeForce RTX 2080Ti. Training a model for 10 epochs on 10,000 audio samples, takes roughly half an hour.

## 4.2 Results

In a first experiment, we evaluate the performance on MFCC features. The results are presented in Table 1. The rows show the respective training sets and the columns the different test set. Gray values

Table 1: **Equal Error Rate (EER) of the baseline classifier on different subset (MFCC).** We train a new GMM model for each training set and use the log-likelihood ratio to score every sample. For each data set we compute the EER, best possible result is 0.0, worst is 1.0, the lower the better. Additionally, we compute the average EER (aEER) over all sets.

| Training Set | LJSpeech | | | | | | JSUT | | |
| | MelGAN | MelGAN (L) | FB-MelGAN | MB-MelGAN | WaveGlow | PWG | MB-MelGAN | PWG | **aEER** |
|---|---|---|---|---|---|---|---|---|---|
| MelGAN | 0.254 | **0.218** | 0.389 | 0.378 | 0.362 | 0.480 | 0.686 | 0.717 | 0.436 |
| MelGAN (L) | **0.286** | 0.126 | 0.402 | 0.347 | 0.345 | 0.478 | 0.456 | 0.492 | 0.364 |
| FB-MelGAN | 0.413 | 0.379 | 0.177 | 0.196 | 0.225 | 0.286 | 0.430 | 0.450 | 0.320 |
| MB-MelGAN | 0.460 | 0.430 | 0.321 | 0.007 | **0.110** | **0.060** | 0.251 | 0.315 | 0.244 |
| WaveGlow | 0.405 | 0.379 | **0.294** | 0.074 | 0.026 | 0.083 | 0.237 | 0.259 | **0.220** |
| PWG | 0.499 | 0.493 | 0.395 | **0.055** | 0.147 | 0.006 | **0.190** | **0.229** | 0.252 |

We highlight in-distribution results in gray and the best out-distribution results per column in **bold**. (L) denotes Large.

Table 2: **Equal Error Rate (EER) of the baseline classifier on different subset (LFCC).** Again, we train a new GMM model for each data set and compute the EER as well as the **aEER**.

| Training Set | LJSpeech | | | | | | JSUT | | |
| | MelGAN | MelGAN (L) | FB-MelGAN | MB-MelGAN | WaveGlow | PWG | MB-MelGAN | PWG | **aEER** |
|---|---|---|---|---|---|---|---|---|---|
| MelGAN | 0.087 | **0.056** | 0.120 | 0.112 | 0.095 | 0.177 | 0.112 | 0.262 | 0.128 |
| MelGAN (L) | **0.082** | 0.024 | 0.089 | 0.092 | 0.079 | 0.162 | 0.142 | 0.370 | 0.130 |
| FB-MelGAN | 0.178 | 0.103 | 0.007 | 0.015 | 0.013 | 0.024 | 0.053 | 0.153 | **0.068** |
| MB-MelGAN | 0.332 | 0.278 | 0.099 | 0.000 | **0.011** | **0.003** | 0.011 | 0.043 | 0.097 |
| WaveGlow | 0.257 | 0.204 | 0.047 | **0.011** | 0.001 | 0.006 | 0.023 | 0.064 | 0.077 |
| PWG | 0.379 | 0.349 | **0.005** | 0.148 | 0.018 | 0.000 | **0.005** | **0.026** | 0.116 |

We highlight in-distribution results in gray and the best out-distribution results per column in **bold**. (L) denotes Large.

indicate that the same generative model is used for the training of the GMM classifier as for the test set.

**MFCC:** When comparing the overall performance, i.e., the lowest average EER (aEER), we can observe that PWG (0.252), MB-MelGAN (0.244), and, WaveGlow (0.220) serve as the best priors for the entire data set. However, they all perform significantly worse on the MelGAN, the MelGAN (L) and (to a lesser extend) the FB-MelGAN data sets. This trend is reversed for MelGAN and MelGAN (L), where they achieve the best results on each other (0.218 and 0.286, respectively) and dropping performance on other data sets ($\sim 0.400$; up to 0.717 on JSUT). FB-MelGAN does not perform particularly well on any data set.

The similarities between PWG and WaveGlow are intuitive. The WaveGlow architecture is heavily inspired by WaveNet (the generator network of PWG). Yet, the best results for both PWG (0.060) and WaveGlow (0.110) are obtained by the model trained on MB-MelGAN. We hypothesize that the auxiliary loss computed over sub-bands forces MB-MelGAN to generate samples more in line with WaveGlow and PWG. Surprisingly FB-MelGAN, generalizes neither to the MelGAN (L) data sets nor to MB-MelGAN. FB-MelGAN uses the same architecture as MelGAN (L) and a similar auxiliary loss to MB-MelGAN, albeit not computing it over sub-bands.

When examining completely novel data (JSUT), all classifier drop in performance. However, PWG, WaveGlow, and, MB-MelGAN still serve as a good prior, implying that the generating architectures exhibit common patterns across different training data sets. A similar pattern was also observed in the image domain [83].

**LFCC:** For comparison we train an additional batch of models on LFCC features. The results can be found in Table 2. LFCC features seem to be a strictly better feature representation, improving performance significantly across the board. Additionally, they allow the classifier trained on FB-MelGAN to become the best performing classifier (0.068). It strikes a balance between generalizing to PWG, WaveGlow, MB-MelGAN, while also retraining a fairly good performance on MelGAN and MelGAN (L). LFCC features contain a significantly higher amount of high-frequency features. Thus, we hypothesize that this fact allows FB-MelGAN to recognize its architecture similarities with MelGAN and the changes caused by the auxiliary loss. Again, similar patterns were also observed in the image domain [21], implying that methods might transfer between the two.

Table 3: **Equal Error Rate (EER) for the baseline classifier in an out-of-distribution setting.** We train a new GMM model for each but one distribution on LFCC features.

| Left-out Set | LJSPEECH | | | | | | JSUT | | |
| | MelGAN | MelGAN (L) | FB-MelGAN | MB-MelGAN | WaveGlow | PWG | MB-MelGAN | PWG | **aEER** |
|---|---|---|---|---|---|---|---|---|---|
| MelGAN | **0.237** | 0.164 | 0.045 | 0.003 | 0.004 | 0.004 | 0.003 | 0.014 | 0.059 |
| MelGAN (L) | 0.233 | **0.166** | 0.037 | 0.002 | 0.004 | 0.002 | 0.002 | 0.014 | 0.058 |
| FB-MelGAN | 0.194 | 0.122 | **0.056** | 0.004 | 0.005 | 0.004 | 0.003 | 0.007 | 0.049 |
| MB-MelGAN | 0.177 | 0.106 | 0.040 | **0.015** | 0.006 | 0.006 | **0.003** | 0.012 | 0.046 |
| WaveGlow | 0.182 | 0.110 | 0.040 | 0.003 | **0.012** | 0.006 | 0.005 | 0.027 | 0.048 |
| PWG | 0.176 | 0.106 | 0.033 | 0.004 | 0.005 | **0.017** | 0.003 | **0.015** | 0.045 |

We highlight the distribution not present in the training set in **bold**. For JSUT, we highlight the entry when the generating network architecture was not part of the training set. (L) denotes Large.

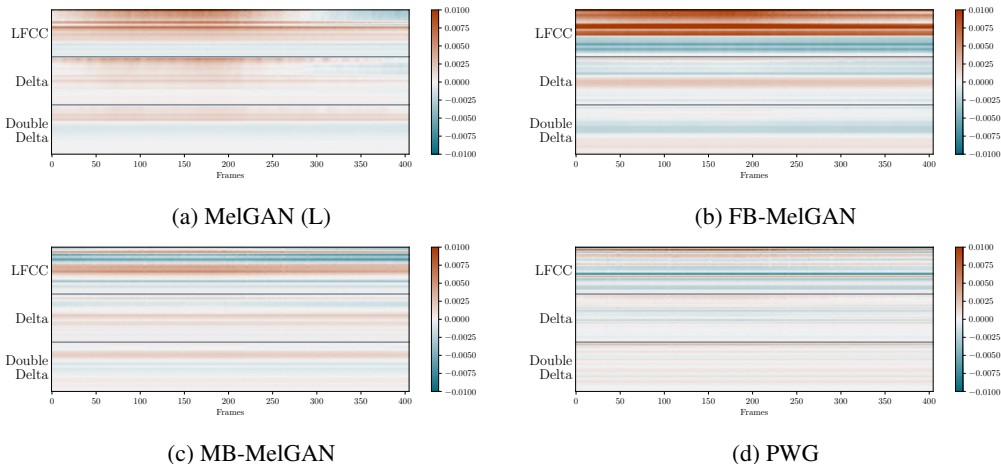

(a) MelGAN (L)

(b) FB-MelGAN

(c) MB-MelGAN

(d) PWG

Figure 3: **Attribution of the different models on a real audio sample.** We show the LFCC, delta, and, double delta features. Since we use a linear filter bank, the plot can be read similarly to the spectrogram plots, i.e., features computed over lower frequencies are at the bottom of their respective plots, features over higher frequencies are at the top. Best viewed in color.

**Leave-one-out:**  Finally, Table 3 present the results of the leave-one-out experiment. We highlight the distribution which was not present in the training data in bold. While we never train on JSUT, we only highlight the distribution if the generating network architecture was not part of the training set. Overall the results improve on the aEER (0.068 → 0.045). Also, the generalization results to a novel setting (JSUT) increase significantly. However, WaveGlow seems to be a key ingredient for good performance on the JSUT-PWG data and the MelGAN and MelGAN (L) data sets still prove to be a challenge, even when included in the training set.

While these first results are encouraging, there is still much room for improvement. Even the best performing classifier trained on multiple network architectures has a false acceptance/false rejection rate of roughly 4.5%.

### 4.3  Attribution

Finally, we want to investigate which parts of the audio signal influence the prediction. To this end, we implemented BlurIG [86], a popular attribution method. We inspect the attribution of four classifier (MelGAN (L), FB-MelGAN, MB-MelGAN and PWG) for the audio clip used in Section 3. The results are displayed in Figure 3, full-sized version are available in the supplementary material. We show the attribution over the LFCC, delta, and, double delta features.

Overall, we can see a shift from very broad attention, spread somewhat evenly across all three feature representations (MelGAN (L)), to a more narrow focused attention across very specific filters (PWG). MelGAN (L) and FB-MelGAN classifiers operate (mostly) on the higher frequencies, while MB-MelGAN and PWG also rely on low frequencies for the detection. These observation confirm our suspicion about the MFCC features. They mask higher frequencies, needed for classifying

MelGAN (L) and FB-MelGAN, while over representing lower frequencies, which still leads to a good performance on the MB-MelGAN and PWG data sets. This also explains the significantly better performance of FB-MelGAN on LFCC features, which strikes a balance between all necessary features. The spread out attribution might also explain the poor in-distribution performance of the classifiers trained on the MelGAN variants, since the classifier needs to focus on a broader range of features.

All in all we can conclude that high frequencies do provide an overall advantage, but lower frequencies cannot be neglected. Thus, we advice that future classifier operate on the entire spectrum.

## 5 Discussion

In this paper we took the first step towards research into audio DeepFakes. While we hope our data set proves useful for future practitioners, there are several limitations to our work:

**Evaluating on realistic data:** The difficulties of obtaining realistic data set has been a long standing problem in the security community [73]. Often benign data is readily available, but data actually used in malicious contexts is hard to come by. This leaves us with estimating real-world performance on proxy data. We argue that in our case, we might have good odds that results transfer. As of right know, images generated by off-the-shelf neural networks are used in malicious attempts [9]. We expect the amount of audio DeepFakes to increase as well.

We also abstain from evaluating a complete TTS pipeline. Completely novel audio is not only influenced by the vocoder but also by the model generating the intermediate representation. While this is an interesting direction for future work, a full evaluation would probably be on the scale of an entire new data set.

An additional line of research is automatic speaker verification, which has been studied in the signal processing community [48, 79, 67]. Due to the similarity of the two domain, we expect that results might transfer between the two. Thus, evaluating models on data sets from both domains, might be beneficial.

**Adversarial examples and perturbations:** DeepFake-image detectors have already been shown to be vulnerable against adversarial examples [10]. There also exists a myriad of adversarial attacks against automatic speech recognition [11, 70, 92, 70, 5, 71, 2] (Abdullah et al. [1] provide a survey). Thus, classifiers should report their robustness against these attacks and common perturbations (noise, room responses, over-the-air settings, etc.) as part of their evaluation. In this work we focused on providing first steps towards audio DeepFake detection. We leave this questions as an interesting direction for future work.

**Variety of the data set:** Our data set presents a first step towards automatic detection of audio DeepFakes. We specifically choose to focus on the LJSPEECH corpus, since it is commonly used for training generative audio models. This allows a one-to-one comparison. However, it only contains recordings by one speaker. While we can make some observation about generalization by comparing against the JSUT data set, a broader analysis focusing on different scenarios would be ideal. We image our corpus being used to study multiple potential classifier designs, evaluating them in a contained environment, before exploring more elaborate settings.

## 6 Conclusion

This paper presents a starting point for researchers who want to investigate generated audio signals. We started by presenting a broad overview of signal processing techniques and common feature representations. Then, we introduced a novel data set, with samples from five different state-of-the-art architectures across two languages. In a first analysis, we already discovered subtle differences between the different models, especially among the higher frequencies. To provide a baseline for future practitioners, we trained several baseline models and evaluated their performance across the different data sets. Finally, we inspected the different classifiers by using an attribution method and found that, while high frequency information proved indispensable, lower frequencies cannot be neglected.

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
