# OpenReview forum: "WaveFake: A Data Set to Facilitate Audio DeepFake Detection"
_NeurIPS.cc/2021/Track/Datasets_and_Benchmarks/Round1 — Submitted to NeurIPS 2021 Datasets and Benchmarks Track (Round 1)_

### Official Review · Reviewer_JAh8 · 2021-06-28
**It’s unclear to me that the proposed dataset can really help audio deepfake detection.**

**Rating:** 5
**Confidence:** 4

**Strengths:**

1. The study of detecting “deepfakes” on the audio/speech domain is timely and can add value to the broader research community.

2. The introduction of an attribution method to reveal that different generative models operate on the low or high frequency looks interesting.


**Weaknesses:**

1. Though the paper claimed to discover the subtle differences between different models (such as operating on lower or higher frequencies), it’s still unclear to me: how the proposed dataset significantly helps detect “deepfakes” on the audio/speech domain. Wouldn’t a dataset that helps detect “deepfakes” be very similar to the real data such that current classifiers cannot discriminate? However, the reported EER looks very large on MFCC.

2. The way of composing a dataset from an ensemble of current generative models does not sound novel to me, as many prior works have widely used it to collect data (in other domains).


**Additional Feedback:**

N/A

**Clarity:**

The paper is fairly well written, though there are many typos. For example, “Chintha et al.combined” (line 135), “generated with different based methods” (line 128), “The rows shows” (line 253) etc. Also for Eq. (6) , What is c here? Is N here the same as the window size of a frame?


**Correctness:**

The claim in this work is not well supported by the experiments. See the Weaknesses for details.

**Documentation:**

There is sufficient detail on documentation.

**Ethics:**

There are few or no ethical concerns that warrant further discussion or review.

**Relation To Prior Work:**

It clearly discussed how this work differs from previous contributions.

**Summary And Contributions:**

This work introduces a new audio dataset that consists of 88,600 audio clips generated from five different generative models. In experiments, two Gaussian Mixture Models (GMMs) trained separately on the real data and the introduced data show the difference of various generative models.

---

> ### Author Response · Authors · 2021-07-11
> **Answer Reviewer JAh8**
>
> We thank the reviewer for his time and interest in reviewing our work.
>
> > Though the paper claimed to discover the subtle differences between different models (such as operating on lower or higher frequencies), it’s still unclear to me: how the proposed dataset significantly helps detect “deepfakes” on the audio/speech domain. Wouldn’t a dataset that helps detect “deepfakes” be very similar to the real data such that current classifiers cannot discriminate? However, the reported EER looks very large on MFCC.
>
> The differences are only visible when plotting the frequency spectograms, when listening to the audio, we could not hear any difference. We have added similar reasoning to the paper.
>
> > The way of composing a dataset from an ensemble of current generative models does not sound novel to me, as many prior works have widely used it to collect data (in other domains).
>
> Please notice that we do not claim to have invented a novel collection technique, our data set is simply novel in the fact that there does not exist a data set for DeepFake audio files.
>
> > The paper is fairly well written, though there are many typos. For example, “Chintha et al.combined” (line 135), “generated with different based methods” (line 128), “The rows shows” (line 253) etc. Also for Eq. (6) , What is c here? Is N here the same as the window size of a frame?
>
> Thank you for pointing these out, we have addressed them and generally proof-read the paper. Additionally, we made it clearer that the c refers to the previous equation.

---

### Official Review · Reviewer_xmWm · 2021-07-05
**Review of WaveFake**

**Rating:** 7
**Confidence:** 3
**Correctness:** The authors' claims about the dataset…

**Strengths:**

The authors present the first dataset of generated audio, to be used in the training of DeepFake detectors.  The dataset represents a timely contribution to an under-addressed area.  There is clear value to researchers in having such a dataset in order to hone detection techniques and learn more about the features of deepfaked audio created by different architectures, even in advance of the wide spread of deepfaked audio. The contribution also includes other aspects of the detector pipeline helpful for researchers' future evaluation of their contributions, such as baseline detectors.

**Weaknesses:**

As the authors note in section 5, Discussion, real world data such as data generated by bad actors is hard to come by.  The authors thus focus on generating proxy data and argue that performance on similar tasks for images transfers.  While this all seems correct, I wonder if the authors' task of "recreating" the datasets on which they trained, producing very similar but deepfaked audio clips, means that performance on their dataset will be less likely to transfer.  Some of the primary malicious use cases for deep fakes center on generating novel phrases and sentences -- e.g., creating a clip that resembles a specific person saying something they have never said, such as the authors mention of a case in which a generated voice recording was used to impersonate a specific CEO.  If the CEO never said those phrases, the task of generating audio in that sounds like his voice is more difficult.  Does that make detection easier, and thus transfer of performance more likely? It would be helpful to address this.

It would also be helpful to provide more analysis of features of the dataset itself, in addition to the analyses of the performance of the detectors.

**Additional Feedback:**

Overall this is a novel and helpful paper which deserves consideration. I commend the authors.

**Clarity:**

The paper overall is clear, but could be organized to make its content more readable and better reach its target audience.  I take it that the paper's audience includes the broader NeurIPS community and is not limited to experts in signal processing.  If part of the paper's aim is to encourage research on deepfake audio detection, this makes sense -- a natural audience would be researchers skilled at detecting deepfaked images or video who have not yet tackled audio alone.

Given that, the paper's introduction of basic signal processing terminology in section 2.1 is especially valuable and could be better situated in the flow of the paper.  It would also be helpful to be more consistent with the level of perceived background knowledge expected.  For example, the paper explains Mel spectrograms but not vocoders.  While the paper is of course not responsible for conveying the content of a graduate seminar on signal processing, more could be done to briefly situate the terms used and enable researchers unfamiliar with the field to hit the ground running.

It would also be helpful to distinguish more clearly in the text of the paper between the dataset on which you trained (the original dataset) and the dataset you created (the novel dataset), as this is a source of some sentence-level confusion.

**Documentation:**

The authors provide a short (one page) datasheet.  It would be valuable to expand this datasheet, following the rubric presented by e.g. Datasheets for Datasets ( https://arxiv.org/pdf/1803.09010.pdf ).  Much of this information is contained in the paper, but if the dataset is to be released into the wild, this information should travel along with it in the datasheet.

**Ethics:**

I found the paper's ethics section to be very underdeveloped. The authors conclude that "Thus, we cannot think of an immediate way to misuse our data."  I agree with the authors that the release of the dataset is likely to be a net positive social good and that its value to researchers outweighs other considerations.

However, there is a clear misuse case for any such public dataset + detector pairing, namely that it could negatively contribute to the deepfakes detection "arms race."  Having a deepfakes dataset + detector could allow bad actors interested in  to train and refine their own detector evasion techniques, improving their abilities to promulgate deepfakes undetected.  This is especially worrying in the misuse case of online harassment or other mass communication on social media platforms.

Imagine an arms race between a generator of deepfaked audio and an in-house detector of deepfaked audio or spam at a social media platform.  At the beginning of the race, the detector is able to detect 93% of deepfaked content. If the bad actor creates 10,000 files, 700 of them are undetected.  Then both the detector and the generator of deepfaked audio train on the authors' novel dataset and detector, one improving its detection technique, the other learning from the authors' detector how to better evade detection.  One question to be considered is whether the detector and generator are likely to improve at the same rate.  In the worst case, the social media platform's in-house detector has already trained on similar data from a private dataset, such that it improves little; the generator, however, improves significantly.  This could lead to a higher 'rate of return' on the enterprise of generating deepfaked audio.

**Relation To Prior Work:**

The authors attest that their paper provides the first (publicly released) dataset of deepfaked audio.  This is correct, as far as I can tell, although I await the verdict of other reviewers, and makes a strong case for its novelty.

On additional factor would be helpful in judging its novelty -- more context for the relationship to the ASVspoof datasets. For example, the authors note that ASVspoof introduced a deepfake audio track as part of the contest in 2021, although did not provide training data.  That provides a clear niche for this dataset's existence (good!) but also prompts the question: how well did the best, contest-winning detectors do on the deepfake audio challenge, even without the benefit of a tailored dataset? How does this compare to detectors trained on this dataset? An answer to that question would make the value of the dataset more clear.

**Summary And Contributions:**

In this paper, the authors introduce a novel dataset of generated audio files, created in order to aid the recognition of deepfakes.  Beginning with two existing repositories of audio clips, "phrases from non-fiction books (LJSPEECH)" and "everyday conversational Japanese 24 (JSUT)", the authors extract Mel spectrograms from each clip, then feed the spectrogram to four models: MelGAN, Multi-band MelGAN, Parallel WaveGAN, and WaveGlow, producing 88,600 generated audio clips in total. They also implement two classifiers as baseline models for future research.

---

> ### Author Response · Authors · 2021-07-11
> **Answer Reviewer xmWm**
>
> We thank the reviewer for the detailed review and very engaging thoughts. We have made several changes in response to his review and outlined them below.
>
> > For example, the paper explains Mel spectrograms but not vocoders.
>
> Creating audio files from text with TTS uses a text representation transformed into an intermediate representation such as Mel-spectrograms. This representation is transformed into raw audio with an uttered version of the original text with a second model. This model is usually referred to as vocoder. We have modified the corresponding section and added a figure to better illustrate this.
>
> >Some of the primary malicious use cases for deep fakes center on generating novel phrases and sentences -- e.g., creating a clip that resembles a specific person saying something they have never said, such as the authors mention of a case in which a generated voice recording was used to impersonate a specific CEO.
>
> We do exclusively train on LJSpeech. Thus, the results on JSUT are results on a completely new speaker which recorded in a different environment and language. We have added more details to the paper.
>
> Completely novel audio is not only influenced by the vocoder but also by the model generating the intermediate representation. While we agree that this is an interesting direction to look into, this would be out-of-scope for a first work in this domain. We have added similar reasoning to the paper.
>
> > It would also be helpful to provide more analysis of features of the dataset itself, in addition to the analyses of the performance of the detectors.
>
> In our paper we aimed at the broader picture and thus refrained from fine-grained analysis. We will look into extending this analysis, but would like pointers on what is of interest for the community. Do you have any suggestions?
>
> > An additional factor would be helpful in judging its novelty -- more context for the relationship to the ASVspoof datasets.
>
> Unfortunately, the challenge is still ongoing, and the results will not be presented until September (thus, even after the second submission round of this track). We should have stated this more clearly in our paper and have modified it accordingly. However, our baseline models are directly adapted from the challenge and can thus be compared. We hope that both this data set and the ASVspoof challenge can benefit from each other after the results of the challenges have been published.
>
> > It would also be helpful to distinguish more clearly in the text of the paper between the dataset on which you trained (the original dataset) and the dataset you created (the novel dataset), as this is a source of some sentence-level confusion.
>
> We have updated the paper accordingly.
>
> > The authors provide a short (one page) datasheet. It would be valuable to expand this datasheet, following the rubric presented by e.g. Datasheets for Datasets ( [https://arxiv.org/pdf/1803.09010.pdf](https://arxiv.org/pdf/1803.09010.pdf) ). Much of this information is contained in the paper, but if the dataset is to be released into the wild, this information should travel along with it in the datasheet.
>
> We have updated the data sheet and merged the corresponding parts of the paper into it.
>
> Note that we already follow the Datasheets for Datasets framework, we simply choose to answer the questions in-line. We also skipped the section on preprocessing since we do not perform any.

---

### Official Review · Reviewer_12ZC · 2021-07-05
**Deep fake audio detection dataset and some analysis**

**Rating:** 5
**Confidence:** 3

**Strengths:**

The problem of audio fake detection is challenging and interesting.

The authors present a variety of experiments with different GAN archiectures analyzing their performance on synthetically generated data from the same and different GANs.



**Weaknesses:**

The dataset is fairly small and is primarily based on LJSpeech dataset, which consists of short audio clips narrated by a single speaker. It is not clear if the methods trained to do well on this benchmark would generalize to novel speakers, or ones with different audio characteristics. Some discussion of the strengths and weaknesses to the proposed benchmark is warranted. For example, how does the method compare to training a model on large-scale synthetic datasets released through competitions such as AVSspoof challenge (https://www.asvspoof.org/)?

There is a concern that the model might learn GAN specific models. One possibility would be to try a leave-one-out validation where a model is trained on all the GAN except one and evaluated on the remaining one.

**Additional Feedback:**

None.

**Clarity:**

Yes. I appreciated the overview of audio representation not being an expert in this domain.

**Correctness:**

The approach appears to be correct. It trains a number of existing GANs on publicly available datasets.

**Documentation:**

The authors raise an interesting question about the terms and conditions for GAN generated data. IMO, the GAN samples are derived from the dataset, so should follow and retain the same rights. For example, a trivial generative model is to sample the data.



**Ethics:**

None.

**Relation To Prior Work:**

The paper describes the related work in vision for deep fake detection and existing benchmarks for the tasks. However, it could better motivate the design choices for this dataset. For example, GANs for generating synthetic images typically require a large dataset for realism. Often these could be millions of images of a narrow domain such as faces. Does this limit the sorts of analysis one can do with GANs trained on small datasets? Or are audio GAN techniques fundamentally easier than vision?

**Summary And Contributions:**

The paper presents a benchmark for detecting fake (synthetically generated) audio clips. The dataset is derived from a combination of LJSpeech and JSUT, on which several GANs are trained to obtain synthetic samples. The paper also presents an overview of audio representations, techniques for generating audio, baseline results for detecting fake audio, and analysis of transferability across different GANs. The results indicate that the problem is challenging --- no single model is able to achieve good generalization across different generative models. There is also analysis of which representations work well for this task (e.g., LFCC is better and the choice of filter banks).

---

> ### Author Response · Authors · 2021-07-11
> **Answer Reviewer 12ZC**
>
> We thank the reviewer for this in-depth feedback. We have taken several steps outlined below to address the issues pointed out in the review.
>
> > The dataset is fairly small and is primarily based on LJSpeech dataset, which consists of short audio clips narrated by a single speaker. It is not clear if the methods trained to do well on this benchmark would generalize to novel speakers, or ones with different audio characteristics. Some discussion of the strengths and weaknesses to the proposed benchmark is warranted.
>
> We have added a discussion point variety to the paper. Please note that we exclusively train on LJSpeech. Thus, the results for JSUT are generalization results to an unseen speaker/language/recording environment. However, we imagine our corpus being used to study multiple potential classifier designs, evaluating them in a contained environment. The results could then be used to extrapolate to more elaborate datasets.
>
> > There is a concern that the model might learn GAN specific models. One possibility would be to try a leave-one-out validation where a model is trained on all the GAN except one and evaluated on the remaining one.
>
> We have run a leave-one-out validation and added the results to the paper. Overall the inclusion of multiple classifiers helps the generalization ability to novel data (JSUT). However, the MelGAN data sets still present a challenge.
>
> > The paper describes the related work in vision for deep fake detection and existing benchmarks for the tasks. However, it could better motivate the design choices for this dataset. For example, GANs for generating synthetic images typically require a large dataset for realism. Often these could be millions of images of a narrow domain such as faces.
>
> We specifically choose LJSpeech since it is commonly used for training generative models. We examined every publication mentioned in our related work section published after the release of LJSpeech (2017). 10 out of 22 trained on private data set and 8 used LJSpeech. The remaining ones used different data sets which did not overlap.
>
> Please note that a one-to-one comparison from audio data to image data is difficult. Since one audio file is a recording of varying length, i.e., a time series. Thus, simply comparing the number of files is insufficient.
>
> Additionally, while earlier GAN models are often trained on big datasets, such as LSUN bedrooms (3 million images), newer models often train on way less data. For example, the most recent version of StyleGAN [1] trained on FFHQ (70,000 faces images), AFHQ (15,000 animal faces), MetFaces (1,336 historical paintings), and Beaches (20,155 photographs of beaches).
>
> [1] Karras, T., Aittala, M., Laine, S., Härkönen, E., Hellsten, J., Lehtinen, J., & Aila, T. (2021). Alias-Free Generative Adversarial Networks. _arXiv preprint arXiv:2106.12423.

---

### Author Response · Authors · 2021-07-11
**High-level Changes Summary**

We would like to thank our reviewers for their encouraging feedback and the appreciation of our work. Based on the feedback received we have made several changes to the manuscript. Here we provide a high-level summary and provide more detailed feedback in the specific comments. We have uploaded an updated version of the manuscript with the changes highlighted in orange:

- As suggested by reviewer 12ZC we have conducted an out-of-distribution experiment. Demonstarting that training on multiple distributions can help the classifier generalize to completely novel data (JSUT).

- Several reviewers commented on the ASVspoof challenge. The challenge is still on-going, which we should have mentioned in the original manuscript. The results are not presented until September (after the second submission round of this track). We have added further discussion on how we imagine that the results of this challenge can be used in conjunction with our work.

- Multiple reviewers commented on the presentation of the evaluation. We have made the distinction between our data set and the reference data more clear. Additionally, we added more discussion on how we think our results generalize.

- We have generally revised the manuscript and extended the underdeveloped section pointed out in the reviews. Note that we do not specifically highlight grammar or typo fixes.

---

### Decision · Program_Chairs · 2021-07-26

**Decision:**

Reject

**Comment:**


This paper presents a novel dataset for audio deepfake detection, consisting of clips generated from 5 different generative models.

The reviewers note that the paper would be strengthened -- especially given the focus of this track -- by a more rigorous analysis of the features of the dataset itself and include more in depth documentation of the dataset following the Datasheets rubric. The work would also be strengthened by a deeper engagement with the ethical dimensions of this work. The authors have addressed some of these concerns in their revised manuscript, including a more comprehensive Datasheet and expanding on the ethical dimensions of the work.

Overall reviewers are still split and there is not a clear champion for the paper so I recommend the authors take the time before the next round of submissions for this track to (a) expand on the work to more deeply connect it to real-world detection of deepfakes and (b) include more analysis of the dataset itself. Such additions would address some of the concerns raised by the reviewers that the papers claims are not well supported by empirical evidence.